# Role of visual and non-visual cues in constructing a rotation-invariant representation of heading in parietal cortex

**Adhira Sunkara[1,2]\*, Gregory C DeAngelis[3]†, Dora E Angelaki[2]\*†**

[1]Department of Biomedical Engineering, Washington University in St. Louis, St. Louis, United States; [2]Department of Neuroscience, Baylor College of Medicine, Houston, United States; [3]Department of Brain and Cognitive Sciences, University of Rochester, Rochester, United States

**Abstract** As we navigate through the world, eye and head movements add rotational velocity patterns to the retinal image. When such rotations accompany observer translation, the rotational velocity patterns must be discounted to accurately perceive heading. The conventional view holds that this computation requires efference copies of self-generated eye/head movements. Here we demonstrate that the brain implements an alternative solution in which retinal velocity patterns are themselves used to dissociate translations from rotations. These results reveal a novel role for visual cues in achieving a rotation-invariant representation of heading in the macaque ventral intraparietal area. Specifically, we show that the visual system utilizes both local motion parallax cues and global perspective distortions to estimate heading in the presence of rotations. These findings further suggest that the brain is capable of performing complex computations to infer eye movements and discount their sensory consequences based solely on visual cues.

\*For correspondence: sunkara@ bcm.edu (AS); angelaki@ cabernet.cns.bcm.edu (DEA)

†These authors contributed equally to this work

Competing interests: The authors declare that no competing interests exist.

## Introduction

Retinal images of the environment are altered by self-generated rotations such as eye or head movements. In order to perceive the world accurately, the component of retinal patterns resulting from such rotations needs to be discounted by the visual system. How the brain achieves such a rotation-invariant visual representation of the world remains unclear. Visually guided navigation is an important context in which achieving rotation-invariance is critical for accurate behavior (*Gibson, 1950*; *Warren and Saunders, 1995*; *Grigo and Lappe, 1999*). For example, while walking down a sidewalk and simultaneously looking at a passing car using eye or head rotations, the brain must discount the visual consequences of the self-generated rotations to estimate and maintain one's direction of translation (i.e., heading).

Self-motion results in retinal velocity patterns known as 'optic flow' (*Gibson, 1950*). During translations, the resulting retinal pattern is generally an expansionary or contractionary radial flow field from which the point of zero velocity (Focus of Expansion, FOE) can be used to estimate heading (*Tanaka et al., 1986*; *Warren et al., 1988*; *Duffy and Wurtz, 1995*; *Britten, 2008*). However, eye or head rotations alter this flow pattern such that deciphering heading requires decomposing the resultant optic flow into translational and rotational components (*Figure 1A*). Psychophysical (*Royden et al., 1992*; *Royden, 1994*; *Crowell et al., 1998*) and electrophysiological (*Bradley et al., 1996*; *Page and Duffy, 1999*; *Zhang et al., 2004*) studies have often emphasized the role of non-visual signals, such as efference copies of self-generated eye/head movements,

**eLife digest** When strolling along a path beside a busy street, we can look around without losing our stride. The things we see change as we walk forward, and our view also changes if we turn our head—for example, to look at a passing car. Nevertheless, we can still tell that we are walking in a straight-line because our brain is able to compute the direction in which we are heading by discounting the visual changes caused by rotating our head or eyes.

It remains unclear how the brain gets the information about head and eye movements that it would need to be able to do this. Many researchers had proposed that the brain estimates these rotations by using a copy of the neural signals that are sent to the muscles to move the eyes or head. However, it is possible that the brain can estimate head and eye rotations by directly analyzing the visual information from the eyes. One region of the brain that may contribute to this process is the ventral intraparietal area or 'area VIP' for short.

Sunkara et al. devised an experiment that can help distinguish the effects of visual cues from copies of neural signals sent to the muscles during eye rotations. This involved training monkeys to look at a 3D display of moving dots, which gives the impression of moving through space. Sunkara et al. then measured the electrical signals in area VIP either when the monkey moved its eyes (to follow a moving target), or when the display changed to give the monkey the same visual cues as if it had rotated its eyes, when in fact it had not.

Sunkara et al. found that the electrical signals recorded in area VIP when the monkey was given the illusion of rotating its eyes were similar to the signals recorded when the monkey actually rotated its eyes. This suggests that visual cues play an important role in correcting for the effects of eye rotations and correctly estimating the direction in which we are heading. Further research into the mechanisms behind this neural process could lead to new vision-based treatments for medical disorders that cause people to have balance problems. Similar research could also help to identify ways to improve navigation in automated vehicles, such as driverless cars.

in discounting rotations to estimate heading. Such non-visual signals can represent several different sources of rotation, including eye-in-head ($R_{EH}$), head-on-body ($R_{HB}$), and body-in-world ($R_{BW}$) movements (*Figure 1B*). Critically, retinal image motion is determined by the translation and rotation of the eye relative to the world ($T_{EW}$ and $R_{EW}$, *Figure 1B*), such that extracting heading from optic flow requires compensating for the total rotation of the eye-in-world (where, $R_{EW} = R_{EH} + R_{HB} + R_{BW}$). Therefore, in general, multiple non-visual signals would need to be added to achieve a rotation-invariant estimate of heading, potentially compounding the noise that is associated with each signal (*Gellman and Fletcher, 1992*; *Li and Matin, 1992*; *Crowell et al., 1998*).

Alternatively, rotation-invariance can theoretically be achieved exclusively through visual processing (*Longuet-Higgins and Prazdny, 1980*; *Rieger and Lawton, 1985*). If the brain can use optic flow to directly estimate and discount rotations of the eye-in-world ($R_{EW}$), such mechanisms may provide a complementary and potentially more efficient way to decompose rotations and translations to achieve invariant heading perception. Psychophysical studies have provided evidence that visual cues may play a role in estimating heading in the presence of rotations (*Grigo and Lappe, 1999*; *Li and Warren, 2000*; *Crowell and Andersen, 2001*; *Li and Warren, 2002*, *2004*; *Royden et al., 2006*). However, electrophysiological evidence for the role of visual cues is ambiguous, in part because previous neurophysiological studies either did not include visual controls for eye rotation (*Zhang et al., 2004*), simulated rotations incorrectly (*Bradley et al., 1996*; *Shenoy et al., 1999*, *2002*) or employed insufficient analysis methods (*Bradley et al., 1996*; *Shenoy et al., 1999*, *2002*; *Bremmer et al., 2010*; *Kaminiarz et al., 2014*) (see 'Discussion').

We recorded neural activity from the macaque ventral intraparietal area (VIP) to evaluate the relative roles of visual and non-visual cues in computing heading in the presence of rotations. To elucidate the role of visual cues, we accurately simulated combinations of translations and rotations using visual stimuli containing a variety of cues present during natural self-motion. Our results provide novel evidence that (1) a subpopulation of VIP neurons utilizes visual cues to signal heading in a rotation-invariant fashion and (2) both local motion parallax and global perspective cues present in optic flow contribute to these computations. In addition, we find that visual and non-visual sources of

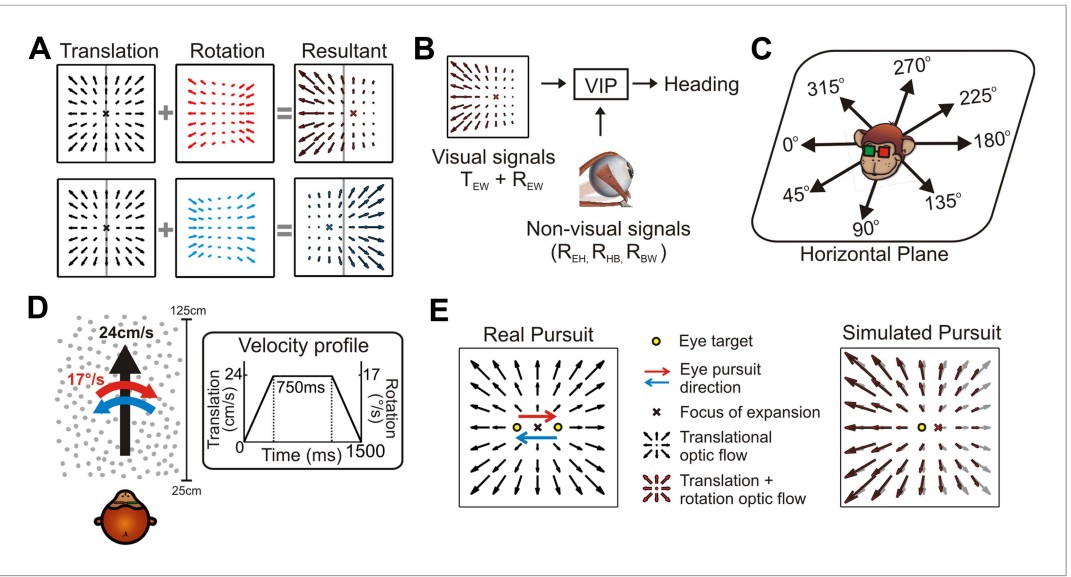

**Figure 1**. The problem of dissociating translations and rotations, and experimental approaches. (**A**) Optic flow patterns during self-motion (shown as planar projections onto a flat image). Forward translations result in symmetric flow patterns (black vector fields) with a focus of expansion (FOE) indicating heading. When rotations are added to forward translations, the resultant optic flow pattern has an FOE shift in the direction of the added rotation (rightward rotation: red, leftward rotation: blue). (**B**) VIP receives both visual and non-visual signals that may be used to achieve rotation-invariant heading estimates. Visual optic flow signals contain information about translation and rotation of the eye in the world ($T_{EW}$, $R_{EW}$) whereas non-visual signals (efference copies) may contain information about rotation of eye-in-head ($R_{EH}$), rotation of head-on-body ($R_{HB}$), or rotation of body-in-world ($R_{BW}$). (**C**) Visual stimuli simulating translations in eight directions spanning the entire horizontal plane were presented to the monkey. (**D**) Schematic showing the translation and rotation parameters in the simulated 3D cloud. Inset shows the trapezoidal velocity profile of translation and rotation during the course of a trial (1500 ms). (**E**) During the 'Real pursuit (RP)' condition, the optic flow stimulus on the screen simulated translation, while rotation was added by having the monkey smoothly pursue a visual target that moved leftward or rightward across the screen. During the 'Simulated pursuit (SP)' condition, the monkey fixated at the center of the display while optic flow simulated combinations of translation and eye rotation. During real and simulated pursuit, the optic flow patterns projected onto the monkey's retina were nearly identical.

The following figure supplement is available for figure 1:

**Figure supplement 1**. Dependence of translational and rotational optic flow properties on viewing distance.

rotation elicit similar responses in VIP, suggesting multi-sensory combination of both visual and non-visual cues in representing rotations. We further show that rotation-invariance is distinct from the reference frame used to represent heading, and provide additional support that heading representation in VIP is close to eye-centered (*Chen et al., 2013*).

## Results

To investigate the effect of rotations on the visual heading tuning of VIP neurons, we presented visual stimuli simulating eight directions of translation in the horizontal plane (*Figure 1C*) and two directions of rotation (*Figure 1D*). To evaluate the relative roles of visual and non-visual cues, rotations were introduced in the form of either 'real' or 'simulated' pursuit eye movements. During real pursuit (RP, *Figure 1E*, left), the monkey smoothly tracked a target moving across the screen such that both visual and non-visual rotation cues were present. During simulated pursuit (SP, *Figure 1E*, right), the visual motion stimulus accurately simulated a combination of translation and eye rotation while the monkey fixated a stationary target at the center of the display (non-visual cues were absent). In order to provide a rich visual environment, the first experiment simulated self-motion through a 3D cloud of dots, a stimulus that contains both local motion parallax cues resulting from translation (*Helmholtz and Southall, 1924*; *Gibson, 1950*; *Longuet-Higgins and Prazdny, 1980*; *Koenderink and van Doorn, 1987*) and global perspective cues to rotation (*Koenderink and van Doorn, 1976*; *Grigo and Lappe, 1999*). To further

explore the underpinnings of a retinal solution in achieving rotation-invariance, a second experiment used a fronto-parallel plane (FP) of dots, which eliminates the local motion parallax cues but retains global perspective cues to rotation.

## Analysis of the effects of rotation on optic flow

When rotation and translation occur simultaneously, the resulting pattern of retinal velocity vectors can differ substantially from the radial optic flow patterns observed during pure translation. This change is often conceptualized as a shift in the focus of expansion (FOE) (*Warren and Hannon, 1990*; *Bradley et al., 1996*; *Shenoy et al., 1999*, *2002*). However, in a visual scene with depth structure, adding rotation results in different FOE shifts at different depths (*Zhang et al., 2004*). This is due to a key difference in the properties of optic flow resulting from translations and rotations—the magnitudes of translational optic flow vectors decrease with distance (depth), whereas rotational optic flow vectors are independent of depth (*Longuet-Higgins and Prazdny, 1980*). Hence, for more distal points in a scene, rotations produce a larger FOE shift (*Figure 1—figure supplement 1*). For the translation and rotation parameters used in this study, the nearest plane in the 3D cloud (25 cm) results in an FOE shift of approximately 20°. However, for any plane farther than 45 cm, the resultant optic flow has an undefined FOE (*Figure 1—figure supplement 1*, top row). The simulated 3D cloud ranged from 25 cm to 125 cm, resulting in a large volume of the stimulus space having undefined FOE shifts. Since FOE shift is an ill-defined measure of the visual consequence of rotations, we simply refer to the net visual stimulation associated with simultaneous translation and rotation as the 'resultant optic flow'.

Forward translations result in an expansionary flow field, for which adding a rightward rotation causes a rightward shift of the focus of expansion (for any given plane). On the other hand, backward translations produce a contractionary flow field and adding a rightward rotation results in a leftward shift in the focus of contraction (*Figure 2A*). If a neuron signals heading regardless of the presence of rotations, then its tuning curves during real and simulated pursuit should be identical to the heading tuning curve measured during pure translation (*Figure 2B*). For a neuron that instead represents the resultant optic flow rather than the translation component (heading), a transformation of the tuning curve is expected due to the added rotations. As a result of the opposite shifts expected for forward (expansionary flow field) and backward translations (contractionary flow field), the heading tuning curve of a neuron preferring forward headings would have a peak that shifts to the right and a trough that shifts to the left during rightward eye rotation; together, these effects cause a skewing of the tuning curve (*Figure 2C*, red curve). For the same neuron, leftward eye rotation would cause the peak to shift to the left and the trough to shift to the right, thus having an opposite effect on the shape of the tuning curve (*Figure 2C*, blue curve). Neurons that prefer lateral headings, which are common in VIP (*Chen et al., 2011*), may in fact, show no shift in the peak. But, since opposite shifts are expected for forward and backward headings, the resulting tuning curve may exhibit substantial bandwidth changes (*Figure 2D*).

Therefore, under the null hypothesis that neural responses are simply determined by the resultant optic flow, the expected effect of rotation on heading tuning is not simply a global shift of the tuning curve, as was assumed previously (*Bradley et al., 1996*; *Page and Duffy, 1999*; *Shenoy et al., 1999*, *2002*; *Bremmer et al., 2010*; *Kaminiarz et al., 2014*). Further illustrations of the expected effects of rotation for hypothetical neurons with different heading preferences are shown in *Figure 2—figure supplement 1*. We designed our quantitative analysis of heading tuning curves specifically to account for these previously unrecognized complexities (see 'Materials and methods').

## Influence of visual and non-visual cues on heading representation in VIP

Heading tuning curves (translation only) can be compared to real pursuit (RP) and simulated pursuit (SP) tuning curves (translation + rotation) to evaluate whether a VIP neuron signals heading invariant to rotations (*Figure 2B*), or whether it responds to the resultant optic flow (*Figure 2C,D*). *Figure 3A* shows heading tuning curves for an example neuron during pure translation (black curve), as well as during rightward (red) and leftward (blue) rotations added in RP and SP conditions. The tuning curves in this example show only minor changes during RP indicating that the cell signals heading in a manner that is largely invariant to eye rotation, consistent with previous findings for real eye rotation (*Zhang et al., 2004*). Interestingly, the tuning curves of the same neuron during SP also change very little, showcasing the role of visual signals in compensating for rotation. Thus, rotation invariance in VIP that was previously attributed to non-visual signals (*Zhang et al., 2004*) might also be driven by visual cues.

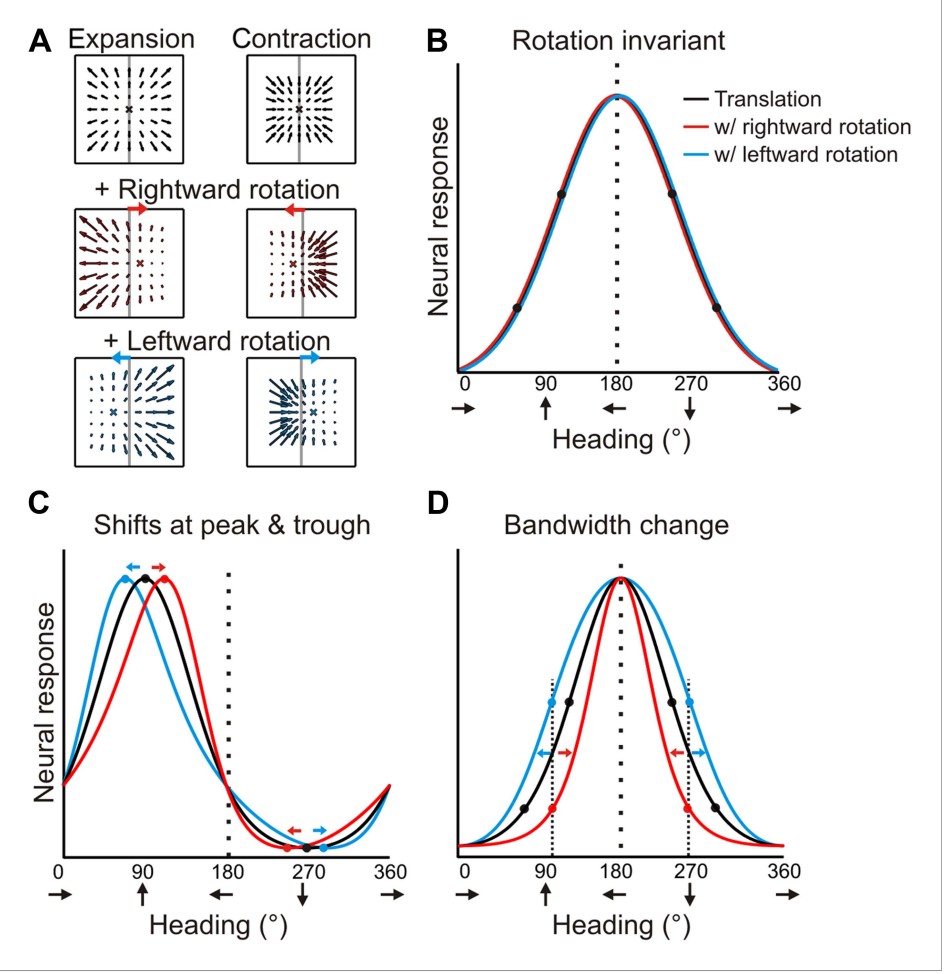

**Figure 2**. Predicted transformations of heading tuning curves due to rotations. (**A**) Forward and backward translations result in expansion and contraction flow fields, respectively (row 1). Adding rotation causes the FOE to shift in opposite directions for forward and backward translations (rows 2, 3). (**B**, **C**, **D**) Hypothetical heading tuning curves showing the predicted transformations due to rotations (rightward, red; leftward, blue). (**B**) Schematic illustration of rotation-invariant heading tuning curves. (**C**) Schematic representing a cell that responds to resultant optic flow (no rotation tolerance) with a heading preference of straight ahead (90°). Rightward rotation causes a rightward shift of the tuning curve for forward headings (around 90°), and a leftward shift for backward headings (around 270°). The opposite pattern holds for leftward rotations. Here, the net result of rotation is a skewing of the tuning curve. (**D**) Schematic tuning of a cell with a leftward heading preference (180°) and no rotation tolerance. In this case, the tuning bandwidth increases for leftward rotations and decreases for rightward rotations. The opposite bandwidth changes would be observed for a cell with a 0° heading preference (see *Figure 2—figure supplement 1*).
The following figure supplement is available for figure 2:

**Figure supplement 1**. Schematic showing tuning curve transformations for hypothetical neurons with different heading preferences.

Data for another example VIP neuron (*Figure 3B*) reveal RP tuning curves that are also largely consistent in shape with the pure translation curve, but which have larger response amplitudes during leftward pursuit. During simulated pursuit, however, the tuning curves of this neuron show clear bandwidth changes. Thus, this second example neuron appears to rely more on non-visual cues to discount rotations. Note that this example neuron preferred lateral headings (leftward) and showed large bandwidth changes during SP, as predicted in the schematic illustration of *Figure 2D*. Such bandwidth changes were observed consistently among VIP neurons that preferred lateral translations; specifically, rightward rotations increased bandwidth for cells preferring rightward headings (~0°) and

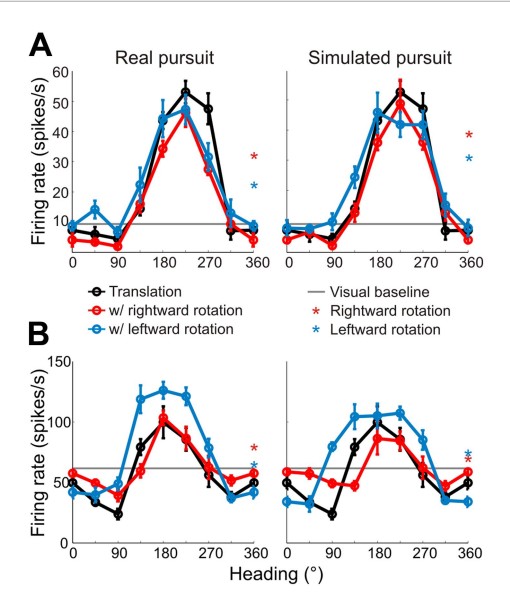

**Figure 3**. Heading tuning curves from two example VIP neurons. Five tuning curves were obtained per cell: one pure translation curve (black), two real pursuit (RP, left column) curves, and two simulated pursuit (SP, right column) curves (rightward rotation: red, leftward rotation: blue). Black horizontal line indicates baseline activity. Red and blue stars in the left column (RP) indicate responses during pursuit in darkness, and in the right column (SP) indicate responses to simulated eye rotation. (**A**) This neuron has largely rotation-invariant tuning curves in both RP and SP conditions (shifts not significantly different from 0, CI from bootstrap), and has significant rotation responses during both pursuit in darkness and simulated rotation (compared to baseline; Wilcoxon signed rank test p < 0.05). (**B**) This example neuron shows significant bandwidth changes during SP (shifts >0°, CI from bootstrap), similar to the prediction of *Figure 2D*. Of the rotation-only conditions, the cell only responds significantly during rightward pursuit in darkness (Wilcoxon signed-rank test p = 0.01).

The following figure supplements are available for figure 3:

**Figure supplement 1**. Bandwidth changes observed in data.

**Figure supplement 2**. Heading tuning curves from two example VIP neurons that preferred forward headings.

decreased bandwidth for cells preferring leftward headings (~180°), with the opposite pattern holding for leftward rotations (*Figure 3—figure supplement 1*). We find analogous results for cells that preferred forward/backward translations. Specifically, we find neurons with tuning curve peaks around forward/backward heading that are invariant to added rotations (*Figure 3—figure supplement 2A*), as well as neurons for which the tuning curve peaks shift with real and simulated pursuit (*Figure 3—figure supplement 2B*), as shown in the simulations in *Figure 2C*.

Because of these changes in tuning curve bandwidth or shape, analysis of the effects of rotation on heading tuning requires more complex and rigorous approaches (*Figure 4—figure supplement 1*) than the cross-correlation or rank-order methods used in previous studies (*Bradley et al., 1996*; *Shenoy et al., 1999*, *2002*; *Bremmer et al., 2010*; *Kaminiarz et al., 2014*). It is also critical to distinguish between changes in response gain and changes in the shape (*Figure 4—figure supplement 2*; see Discussion) of tuning curves, which our analysis allows because we sample the entire heading tuning curve (*Mullette-Gillman et al., 2009*; *Chang and Snyder, 2010*; *Rosenberg and Angelaki, 2014*). As shown in *Figure 4—figure supplement 1*, the first step in the analysis involves normalizing each RP and SP tuning curve to match the dynamic range of the pure translation tuning curve. Following this transformation, the change in the shape of the RP and SP tuning curves can be measured without ambiguity. To account for the expected changes in bandwidth and skew, partial shifts of the tuning curve were measured separately for forward (0°:180°) and backward (180°:360°) headings. Thus, four shift values were obtained from each neuron for both real and simulated pursuit, corresponding to forward/backward headings and left/right rotation directions. These four values were averaged for each neuron to quantify the transformation in shape and obtain one shift metric for RP tuning curves and one for SP tuning curves (see 'Materials and methods', *Figure 4—figure supplement 1*).

Results are summarized for the population of recorded neurons (n = 72; from two monkeys) in

**Figure 4**. A shift of 0° implies that the neuronal representation of translation is invariant to rotation (i.e., the shape of heading tuning curves are highly similar, as in *Figure 3A*). A positive shift indicates under-compensation for rotation, such that responses change in a manner consistent with the resultant optic flow. Negative shifts indicate that the tuning curve transformation was in the direction opposite to that expected based on the resultant optic flow. This can be interpreted as an over-compensation for rotation. As noted earlier, though the FOE shift for the nearest depth plane (25 cm) in our stimuli is 20°, a majority of the cloud volume (45–125 cm deep) is dominated by rotations, such

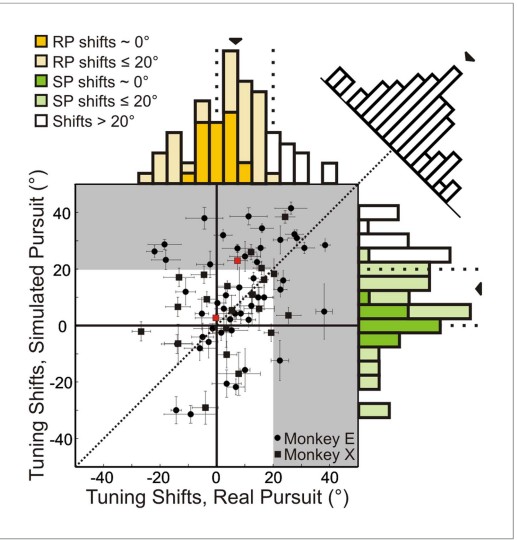

**Figure 4**. Scatterplot and marginal distributions of shifts measured during real pursuit (RP) and simulated pursuit (SP) using 3D cloud stimuli (n = 72 cells). A shift of 0° indicates rotation-invariance. Positive and negative shifts indicate under-compensation and over-compensation for rotation, respectively. Grey shaded area corresponds to shifts >20° (conservative estimate of shift for cells with no tolerance to rotations). Red data points correspond to the shifts associated with the example cells shown in *Figure 3*. Error bars depict bootstrapped 95% confidence intervals (CI). Colored regions of marginal distributions indicate shifts ≤20°. Darker colors indicate shifts not significantly different from 0°. Uncolored histograms indicate shifts significantly >20°. Diagonal histogram shows difference in RP and SP shifts for each neuron with a median of −6.0° indicating that for most cells SP shifts tended to be larger than RP shifts (significantly <0°; Wilcoxon signed-rank test p = 0.02).

The following figure supplements are available for figure 4:

**Figure supplement 1**. Method for analyzing tuning curve shifts.

**Figure supplement 2**. Problems with previous approaches to measuring shifts in the absence of full tuning curve measurements.

that the resultant optic flow has undefined FOEs. This implies that neurons should show shifts that are generally much larger than 20° if they do not discount the rotations and merely represent the resultant optic flow.

In the RP condition, 22/72 (30.6%) neurons showed shifts that were not significantly different from zero (bootstrap 95% CI); these cells can be considered to represent heading in a rotation-invariant fashion. For SP, 17/72 (23.6%) neurons had shifts that were not significantly different from zero, indicating that purely visual cues were sufficient to achieve rotation-invariance in these neurons. Only 13/72 (18.1%) neurons during RP and 19/72 (23.4%) neurons during SP showed shifts that were significantly greater than 20°, suggesting that only a minority of VIP neurons simply represent the resultant optic flow.

The median shift of the population during RP is 8.5°, which is significantly less than the 13.8° median shift observed during SP (Wilcoxon signed-rank test; p = 0.02), indicating greater tolerance to rotations in the presence of both non-visual and visual cues. However, both median shifts are significantly greater than 0° (Wilcoxon signed-rank test; p < 0.001), and less than 20° (Wilcoxon signed-rank test; RP: p < 0.001, SP: p = 0.005) suggesting that, on average, VIP neurons do not simply represent the resultant optic flow, but rather signal heading in a manner that is at least partially tolerant to rotations. Together, these findings indicate that VIP can signal heading in the presence of rotations using both visual and non-visual cues. Importantly, this tolerance to rotations is observed even when only visual cues are present (SP).

## Visual and non-visual rotation signals in VIP

The previous section shows that VIP neurons can use visual cues to signal heading in the presence of rotations, but it is unclear if the rotational component is also represented. During real pursuit, the rotation arises from a movement of the eye relative to the head. In this case, both non-visual and visual sources of information about the rotation are available. These two sources of information differ in that the non-visual source signals the rotation of the eye relative to the head ($R_{EH}$) and the visual source signals the rotation of the eye relative to the world ($R_{EW}$). Previous studies have shown that VIP receives efference copies of pursuit eye movements (*Colby et al., 1993*; *Duhamel et al., 1997*), reflecting an $R_{EH}$ signal. However, no previous studies have tested if VIP also carries an $R_{EW}$ signal based on visual rotation information present in optic flow.

To test whether neurons in VIP signal rotations based on both non-visual and visual cues, we analyzed data from interleaved rotation-only trials (leftward and rightward rotations) in which the monkey either pursued a target in darkness (non-visual $R_{EH}$ signal) or fixated centrally while the visual stimulus simulated a rotation (visual $R_{EW}$ signal) with the same velocity profile as pursuit in darkness. We found that about half of the rotation responses were significantly different from baseline activity

during both real and simulated rotations (144 responses from 72 cells; 73/144, 50.7% during pursuit in darkness and 78/144, 54.2% during simulated rotation). Since we only tested horizontal (yaw axis) rotations at a single constant velocity, it is likely that more VIP neurons are responsive to rotation, but prefer different rotation velocities or axes of rotation.

In our experiments, the $R_{EW}$ signal is equivalent to the $R_{EH}$ signal since only eye rotations are considered. Therefore, similarity between the efference copy signal ($R_{EH}$) and the neural responses to purely visual rotation stimuli ($R_{EW}$) would suggest the presence of an integrated (visual and non-visual) $R_{EW}$ signal in VIP. We find that the baseline-subtracted responses to these two types of rotation stimuli are significantly correlated (rightward rotation: Spearman r = 0.50, p < 0.001; leftward rotation: Spearman r = 0.39; p = 0.001), supporting the presence of a rotation signal derived from purely visual cues ($R_{EW}$) in area VIP (*Figure 5A*). Furthermore, the difference in response between rightward and leftward rotations (*Figure 5B*) shows that many VIP neurons exhibit direction-selective responses to rotation. We also find significant correlation between the differential responses (left—right rotation) during real and simulated rotation (Spearman r = 0.59; p < 0.001). These results support the hypothesis of multi-sensory convergence of visual and non-visual cues to provide consistent rotation information, which may be critical for encoding rotations, in addition to achieving a rotation-invariant representation of heading.

It is important to note that, in general, retinal motion corresponding to $R_{EW}$ is a combination of $R_{EH}$, rotation of the head-on-body ($R_{HB}$), and body-in-world ($R_{BW}$). And each of these different rotations will be accompanied by different efference copy signals. If VIP neurons represent $R_{EW}$ based on non-visual signals, then they would have to represent a combination of all efference copy signals: $R_{EW} = R_{EH} + R_{HB} + R_{BW}$. Although we cannot test this directly with our data, the correlations observed in *Figure 5* allow for the possibility that VIP neurons represent $R_{EW}$ based on both visual and non-visual cues.

## Role of perspective distortions in achieving rotation-invariance

Results from the 3D cloud experiment (*Figure 4*) demonstrate, for the first time at the neural level, a clear contribution of visual cues in achieving a rotation-tolerant representation of heading. To gain a deeper understanding of the visual mechanisms involved in dissociating translations and rotations,

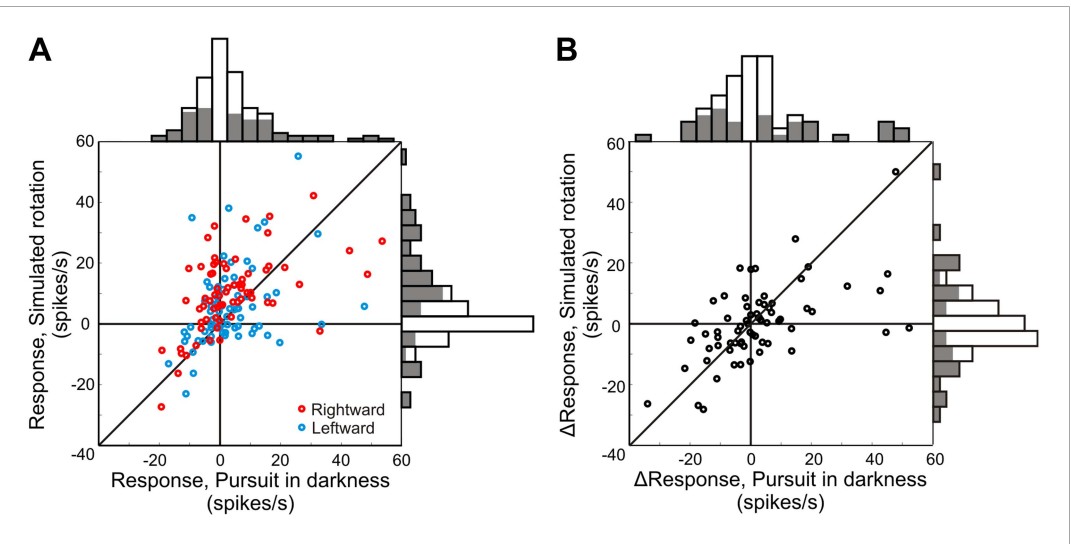

**Figure 5**. Neural responses to pure rotation stimuli. (**A**) Scatterplot and marginal distributions of baseline-subtracted rotation responses. The monkey either pursued a target across a dark screen (pursuit in darkness) or fixated centrally as rotation was simulated in the 3D dot cloud (simulated rotation). Filled marginal distributions indicate significant rotation responses compared to baseline (t-test, p ≤ 0.05). Red and blue symbols denote rightward and leftward rotations, respectively. (**B**) Scatterplot of differences between leftward and rightward rotation responses. Filled marginal distributions indicate significant differences between leftward and rightward rotation responses (t-test, p ≤ 0.05).

we investigated which optic flow properties are used by the visual system to infer self-motion from visual cues. *Helmholtz and Southall (1924)* and *Gibson (1950)* suggested that local motion parallax information plays an important role in deciphering self-motion based on the depth structure of a scene. In a 3D environment, two points can have similar retinal locations, but different depths. As illustrated in *Figure 1—figure supplement 1*, optic flow vectors resulting from observer translation are dependent on depth, producing different retinal velocities for points at different depths. This difference in velocity between nearby points at different depths gives rise to local motion parallax. Rotations, on the other hand, produce image motion that is not depth-dependent, and therefore lacking local motion parallax. As a result, for a rich 3D environment, computing the local difference between optic flow vectors corresponding to points at different depths allows the rotational component of optic flow to be subtracted away (*Longuet-Higgins and Prazdny, 1980*; *Rieger and Lawton, 1985*; *Warren and Hannon, 1990*), and the singularity point of the resulting motion parallax field (*Figure 6A*) corresponds to the observer's heading. This solution requires rich depth structure in the scene, which is not always present. For instance, walking through a dense forest provides robust local motion parallax cues, but walking towards a wall or through an open field, does not.

In addition to local motion parallax cues resulting from observer translation, optic flow also contains global components of motion that convey information about observer rotation. When a pure eye rotation is simulated using optic flow stimuli, the image contains distortions resulting from the changing orientation of the eye relative to the scene, that we term 'dynamic perspective cues' (see *Kim et al., 2014* for more details). A correct simulation of rotational optic flow can thus be characterized as a combination of laminar flow and dynamic perspective cues (*Figure 6B*). Importantly, these cues are independent of the depth structure of the scene and are present in scenes having rich 3D structure as well as scenes consisting of a single plane. Theoretical studies have proposed that such cues may play an important role in estimating and discounting the rotational component of optic flow to estimate heading (*Koenderink and van Doorn, 1976*, *1981*; *Grigo and Lappe, 1999*). A recent electrophysiological study in MT provides evidence that the visual system may be capable of using these dynamic perspective cues to disambiguate the sign of depth specified by motion parallax (*Kim et al., 2014*).

To examine the role of dynamic perspective cues, we conducted a second set of experiments using a fronto-parallel (FP) plane of dots with zero disparity. These visual stimuli contain global perspective cues to rotation, as in the 3D cloud stimulus, but lack local motion parallax cues. For 11/34 neurons recorded, the stimulus was viewed binocularly; the remaining cells were recorded while the monkey viewed the stimulus monocularly with the eye contralateral to the recording hemisphere. In contrast to previous studies, which kept the simulated distance to a FP wall constant over the duration of a trial (*Bradley et al., 1996*; *Shenoy et al., 1999*, *2002*), the simulated distance of the FP plane changed, in our stimuli, from 45 cm at the beginning to 18 cm at the end of the trial. This more accurately simulates the real world situation in which approaching a wall reduces its distance from the observer over time. As a result, the speed of the translation component of optic flow increased over time for forward translations as the distance to the wall decreased (*Figure 1—figure supplement 1*). Since rotational optic flow is invariant to the distance from a wall (*Figure 1—figure supplement 1*), the resulting shift in FOE due to added rotations changed over time in our stimulus. During the middle 750 ms of a forward translation stimulus, real or simulated pursuit results in an average FOE shift of 37°. Hence, heading tuning shifts significantly smaller than 37° would provide evidence for the hypothesis that the visual system can use dynamic perspective cues to discount rotations.

*Figure 6C* summarizes the shifts in heading tuning measured during presentation of the FP plane stimulus. The median shifts across the population for real pursuit (14.3°) and simulated pursuit (21.5°) were both significantly less than the 37° expected if there were no tolerance for rotations (Wilcoxon signed-rank test; $p < 0.005$). The median values were also significantly different from each other (Wilcoxon signed-rank test; $p = 0.03$) and greater than 0° (Wilcoxon signed-rank test; $p < 0.001$). Furthermore, 8/34 (23.5%) neurons during RP and 5/34 (14.7%) neurons during SP had shifts that were not significantly different from 0° (darker colors in *Figure 6C*), implying rotation-invariant heading responses. Only 6/34 (17.6%) neurons during RP and 12/34 (35.3%) neurons during SP showed shifts that were statistically greater than or not different from 37° (95% CI; see 'Materials and methods'). These results indicate that, even in the absence of non-visual signals and 3D visual cues such as local motion parallax, a large sub-population of VIP neurons can use global perspective cues to at least partially mitigate the effect of rotations on heading tuning. Shifts measured during simulated pursuit in the 3D cloud experiments were significantly less than shifts

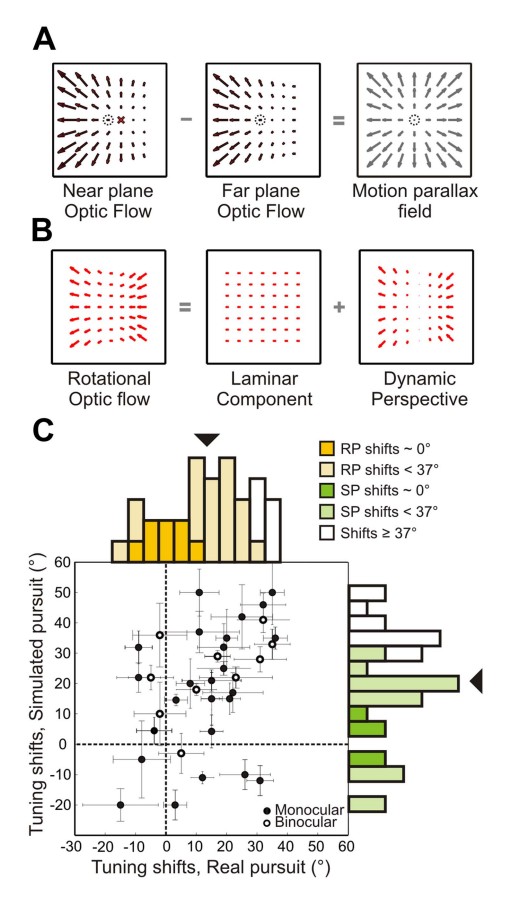

**Figure 6**. Role of dynamic perspective cues in signaling rotation-invariant heading. (**A**) Optic flow fields during combined translation and rotation at two different depth planes have different FOE shifts. The dotted circle indicates true heading. Subtracting these flow fields yields a motion parallax field that eliminates the rotational component and the point of zero local motion parallax corresponds to the true heading. (**B**) Rotational optic flow can be decomposed into laminar flow and dynamic perspective cues. Dynamic perspective cues may signal eye rotations even in the absence of depth structure. (**C**) Scatterplot and marginal distributions of shifts measured using the fronto-parallel plane stimulus during real and simulated pursuit (n = 34 cells). Format as in *Figure 4*. Open and filled symbols denote data collected during binocular and monocular viewing, respectively. Errorbars denote bootstrapped 95% CIs. All filled histograms indicate shifts significantly <37°. Dark colored histogram bins indicate cells with shifts not significantly different from 0°. Uncolored bars indicate shifts ≥37°.

measured using the FP plane (Wilcoxon rank sum test; p = 0.02). This implies that both local motion parallax cues arising from translations, and global features such as dynamic perspective cues arising from rotations play important roles in visually dissociating translations and rotations.

## Reference frames for representing heading

Since the eyes physically rotate during real pursuit, but the head does not, previous studies interpreted rotation-invariant heading tuning as evidence that VIP neurons represent self-motion in a head-centered reference frame (*Zhang et al., 2004*). In contrast, studies that measured heading tuning with the eye and head at different static positions have revealed an eye-centered reference frame for visual heading tuning in VIP (*Chen et al., 2013*, *2014*). On the surface, these results appear to be incompatible with each other. However, we posit that the issues of rotation-invariant heading tuning and reference frames are not necessarily linked. Indeed, we show below that VIP neurons can discount rotations without signaling translation direction in a head-centered reference frame.

The key to reconciling these issues is appreciating that, during eye pursuit, the eye-centered reference frame rotates relative to a subject's heading (*Figure 7A*). As the eye rotates, the direction of translation remains constant in head-centered coordinates (*Figure 7A*, dashed green lines). However, in the rotating eye-centered reference frame, the translation direction relative to the eye changes over time, such that the focus of expansion moves across the retina (*Figure 7B*). This is true for both the RP and SP conditions. During RP, the eye physically moves and the FOE remains constant on the screen, whereas during SP, the eye remains stationary as the FOE drifts across the screen. Hence, the temporal change in the translation direction with respect to the retina is the same during both real and simulated pursuit.

In our experimental protocol, as well as that of previous studies (*Bradley et al., 1996*; *Shenoy et al., 1999*, *2002*; *Zhang et al., 2004*), the average eye position during the translation-only, real pursuit and simulated pursuit conditions is the same (centered on the screen) over the duration of a trial. Therefore, the average eye position is the same as the average head position. As a result, time-averaged neural responses may provide insight into what signal is represented (heading or resultant optic flow), but not about whether these signals are represented in an eye- or head-centered reference frame. To evaluate reference frames, responses must be examined with the eye at different positions relative to the simulated heading. In our experiments, we

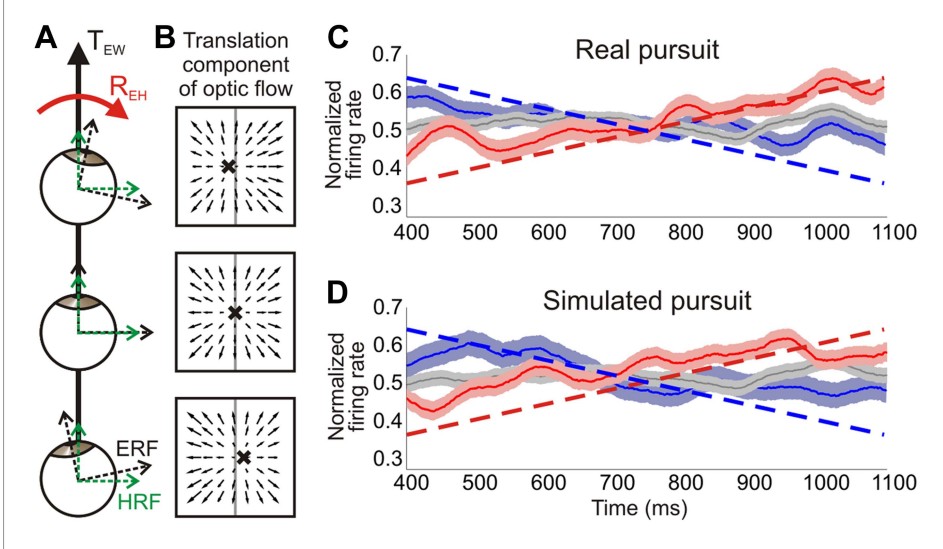

**Figure 7**. Distinguishing reference frames from rotation invariance. (**A**) Schematic of a rightward eye rotation while translating forward. As the eye position changes during smooth pursuit, the eye reference frame (ERF, black axes) rotates relative to the head ($R_{EH}$) and the direction of translation in the world, $T_{EW}$. Since the head is not rotating relative to the world, the head reference frame (HRF, green axes) remains constant with respect to the heading. (**B**) In retinal co-ordinates, the translation component of optic flow changes with eye position and results in a drifting FOE (x) across the retina. The translation direction represented by the FOE changes from right of straight ahead to left of straight ahead for rightward rotations. (**C, D**) Heading corresponding to the largest firing rate gradient was identified for each neuronal tuning curve and the temporal responses at that heading were evaluated. Dashed straight lines show the predicted population response slopes based on the assumption of an eye-centered reference frame. The population average of the normalized time course of firing rate is plotted for each condition type—translation only (grey), rightward rotation (red) and leftward rotation (blue) for real pursuit (**C**) and simulated pursuit (**D**). Shaded regions indicate standard errors. The significant positive and negative trends observed are consistent with a reference frame that is intermediate between eye- and head-centered, but closer to an eye-centered reference frame.

can examine the temporal responses of neurons to study reference frames since the translation direction in eye coordinates changes over time. Accordingly, an eye-centered representation of heading would result in systematic temporal response variations due to the rotating reference frame, and these variations would be different for leftward and rightward rotations of the eye. In contrast, a head-centered representation would result in responses that are constant over time, and similar for rightward and leftward rotations during both real and simulated pursuit.

For a neuron representing heading in an eye-centered reference frame, a rightward eye rotation would result in an upward trend in firing rate over time for headings along the positive slope of the tuning curve. In contrast, a leftward eye rotation would result in a downward trend (*Figure 7C,D*, dashed lines). It is important to note that these trends are determined by the changing eye position and are independent of how tolerant the heading representation is to rotations (i.e., the extent of compensation). The degree of rotation compensation would result in a shift in the mean firing rate away from the pure translation responses (as discussed in previous sections), irrespective of the reference frame in which translations are represented. Therefore, neurons can represent translations invariant to rotations in either a head-centered or an eye-centered reference frame.

In order to evaluate the underlying reference frame for representing translations in area VIP, we examined the temporal changes in firing rate for each neuron over the same 750 ms epoch used in the rest of the analyses. If neurons signal heading in an eye-centered reference frame, the largest temporal variations in firing rate will occur at headings along the steepest portion of the tuning curve. Therefore, we identified the heading corresponding to the largest positive gradient for each tuning curve, and examined the temporal dynamics of responses for that direction. In order to determine the expected temporal changes in firing rate under the assumption of an eye-centered reference frame,

the slope of the tuning curve at the heading corresponding to the largest gradient was calculated for each normalized tuning curve. The average predicted slopes for the population based on our data were $\pm 0.41$/s during real pursuit and $\pm 0.4$/s for simulated pursuit (dashed lines in *Figure 7C, D*).

These predictions, based on an eye-centered reference frame hypothesis, were compared to the average time course of normalized responses of the population of VIP neurons (see 'Materials and methods' for details). VIP population responses show trends in the directions predicted by an eye-centered reference frame, but are inconsistent with the expectation for a head-centered reference frame (red, blue curves in *Figure 7C,D*). The slopes observed in VIP responses correspond to an intermediate reference frame that lies closer to an eye-centered frame than a head-centered reference frame. Specifically, for real pursuit, average responses increased for rightward eye rotation (slope = 0.29/s, 95% CI = [0.21 0.37], linear regression) and decreased for leftward rotation (slope = $-0.24$/s, 95% CI = [$-0.16$–0.32]). These slopes are significantly different from 0 and ~65% as steep as the predictions of the eye-centered reference frame, thus indicating an intermediate reference frame. Since the temporal response profile was essentially flat during the translation only condition (slope = 0.01/s, 95% CI = [$-0.04$ 0.06]) and opposite trends are observed for rightward vs leftward rotations, these response patterns cannot be explained by other basic aspects of neural response dynamics, such as adaptation.

Interestingly, similar trends are also observed during simulated pursuit (rightward: slope = 0.28/s, 95% CI = [0.21 0.35]; leftward: slope = $-0.26$/s, 95% CI = [$-0.17$–0.35], linear regression), for which the eye does not physically rotate. These slopes are again about two-thirds as steep as expected based on the eye-centered reference frame hypothesis. Whereas previous studies have demonstrated a role of non-visual signals in estimating the position of the eye or head relative to the body (*Squatrito and Maioli, 1997*; *Lewis et al., 1998*; *Klier et al., 2005*), these results suggest that visual signals in VIP carry information about changes in eye position even in the absence of efference copy signals. In other words, the temporal dynamics of an eye rotation may be inferred from the rotational components of optic flow and used to modulate neural responses during simulated pursuit. This further strengthens the functional role of visual signals in VIP for estimating rotational information and contributing to a rotation-invariant heading representation.

## Discussion

We evaluated how heading is represented in macaque area VIP in the presence of rotations, finding that a sub-population of VIP neurons represent heading in a rotation-invariant fashion while a majority of the population is at least partially tolerant to rotations. Importantly, rotation invariance can be achieved using both non-visual and purely visual cues. Previous neurophysiology literature emphasized the importance of non-visual cues, especially efference copy signals, but clear evidence for the role of visual cues has been missing, as discussed below. In contrast, our study provides novel evidence for the role of visual cues in discounting rotations and representing heading. Furthermore, we show that both local motion parallax and global dynamic perspective visual cues present in optic flow play a significant role in decomposing the components of self-motion. The importance of visual signals is reinforced by our finding that VIP neurons also carry rotation signals derived from purely visual cues. The significant correlation between visual and non-visual rotation responses is consistent with a multi-sensory representation of rotations. In addition, we resolve an important ambiguity in the literature between the concepts of tolerance to rotations and reference frames. Specifically, we examine the effect of a rotating eye reference frame on visual responses to show that rotation tolerance does not necessarily imply a head-centered reference frame. Our findings show conclusively that visual cues play a significant role in achieving rotation-invariant heading representations.

### Importance of visual cues

It is important to recognize that the significance of visual cues in discounting rotation extends beyond eye pursuit to head-on-body ($R_{HB}$) and body-in-world ($R_{BW}$) rotations as well. The efference copy signals for each of these sources of rotation depend on the specific motor commands generating the movement. If we consider that eye, head, and body rotations are often generated simultaneously, multiple efference copy signals must be added together and subsequently discounted from the resultant optic flow to signal heading accurately. Each of these non-visual signals is associated with signal-dependent noise (*Gellman and Fletcher, 1992*; *Li and Matin, 1992*; *Crowell et al., 1998*); thus, combining multiple, potentially independent, efference copy signals to estimate rotations may

not always be an efficient solution for the brain. On the other hand, the information contained in visual cues is independent of the source of rotation and represents rotation of the eye relative to the world ($R_{EW}$). The $R_{EW}$ information present in optic flow inherently reflects the sum of all the different sources of rotation ($R_{EW} = R_{EH} + R_{HB} + R_{BW}$) and thus provides direct information regarding the total rotation of the eyes during self-motion. Therefore, visual signals may have important advantages when the goal is to accurately estimate heading in the presence of self-generated rotations.

However, we also face situations in which visual information may be sparse, such as driving at night on an open road (limited visual range and depth structure), where non-visual signals may be crucial. As expected, given the brain's propensity towards multi-sensory integration, we find that both visual and non-visual signals contribute to discounting rotations to represent heading. Real pursuit shifts are smaller than simulated pursuit shifts, and both types of shifts are smaller for a dense 3D cloud than a fronto-parallel plane.

Given the variety of efference copy signals present in parietal cortex (*Andersen, 1997*) and the correlation observed between the $R_{EH}$ (pursuit in darkness) and $R_{EW}$ (pure simulated rotation) responses in our data (*Figure 5*), we postulate that VIP contains an integrated representation of rotation that relies on both visual signals and efference copy inputs. However, to conclusively test these theories, experiments with multiple rotation velocities and directions as well as different sources of rotation (e.g., eye vs head pursuit) need to be conducted. How these visual rotation cues are combined with efference copy signals and other non-visual sensory cues to rotation (e.g., vestibular inputs) warrants further investigation.

## Comparison to previous behavioral studies

Several human psychophysical studies have assessed pursuit compensation during heading estimation based on visual and non-visual cues. However, owing to variations in experimental protocols, visual stimuli, and instructions given to the subjects, the results of these studies vary substantially. If we consider studies that used 3D cloud stimuli, we find that some studies report large errors in heading perception (the difference between reported heading and true heading) in the absence of efference copy signals (*Royden et al., 1992*; *Royden, 1994*; *Banks et al., 1996*), whereas other studies report that subjects are able to accurately perceive their heading based on purely visual stimuli (*Warren and Hannon, 1988*, *1990*; *van den Berg and Brenner, 1994*). In order to compare results across these studies, we calculated the degree of compensation as the difference between the error in heading perception and the shift in FOE based on the experimental parameters, normalized by the expected shift in FOE ([FOE shift-heading error]/FOE shift). The rotation compensation observed in these studies during simulated pursuit (only visual cues) ranged from 100% to 20% for a 3D cloud stimulus (based on the depth plane corresponding to the screen distance). Studies with smaller compensatory effects (*Royden et al., 1992*; *Royden, 1994*; *Banks et al., 1996*) concluded that optic flow was insufficient for estimating translations in the presence of rotations. However, these studies used visual stimuli with a small field of view and a low density of dots in the 3D cloud, thus limiting the visual information available for estimating heading in the presence of rotations (*Grigo and Lappe, 1999*). Despite these limitations in the visual stimuli, the compensatory effects were greater than 0. Moreover, other studies have shown that richer visual stimuli, including landmarks (*Li and Warren, 2000*, *2004*; *Royden et al., 2006*) and larger fields of view (*van den Berg and Brenner, 1994*; *Grigo and Lappe, 1999*), resulted in larger compensatory effects based on purely visual cues.

In this study, using a 3D cloud stimulus, we observed a large and continuous range of compensatory effects, including a substantial subset of VIP neurons that compensated completely for rotations, as well as neurons that do not compensate at all or even over-compensate for rotations. Since the experimental parameters used in our study and the various behavioral papers are different, it is difficult to compare our results quantitatively with the published behavioral findings. However, the fact that we find moderate, but significant, compensation during simulated pursuit is broadly consistent with the psychophysical literature.

Furthermore, depending on how the population of VIP neurons is decoded, a substantial range of behavioral effects might be expected. For instance, if the rotation-invariant neurons are selectively decoded to estimate heading, it should be possible for VIP to support behavioral responses with compensation close to 100%. On the other hand, if all VIP neurons are decoded with equal weights, we would expect the behavioral errors to be comparable to the mean compensation observed in the neural population. It is also important to note that in many behavioral studies, subjects made small but

significant errors even during real pursuit (*Freeman, 1999*; *Freeman et al., 2000*; *Crowell and Andersen, 2001*), consistent with our finding that the average compensation among VIP neurons is not complete even when both visual and non-visual cues to rotation are available.

Some psychophysical studies attribute the errors observed during simulated pursuit to the misinterpretation of path-independent rotations (such as eye pursuit during straight translations) as motion along a curved path (*Royden, 1994*; *Royden et al., 2006*). In behavioral studies that eliminate this ambiguity through specific instruction to subjects, heading errors during simulated pursuit are reported to be largely reduced (*Li and Warren, 2004*; *Royden et al., 2006*). This provides further evidence that the visual system is indeed capable of estimating rotation-invariant heading based on purely visual stimuli. It is also possible that the range of compensation observed in our data could be a result of this perceptual ambiguity. To evaluate how the brain resolves this ambiguity, neurophysiological studies using both path-independent rotations and curved path stimuli are needed.

## Comparison with previous electrophysiological studies

Previous physiological studies emphasized the contribution of efference copy signals to achieving rotation invariance (*Bradley et al., 1996*; *Page and Duffy, 1999*; *Shenoy et al., 1999*; *Zhang et al., 2004*). However, these studies could not conclusively establish a contribution of visual rotation cues to heading tuning for various reasons. Some studies did not use a simulated pursuit condition and therefore could not disambiguate visual and non-visual contributions to the rotation-invariance of heading tuning they observed (*Page and Duffy, 1999*; *Zhang et al., 2004*). On the other hand, *Bradley et al. (1996)* and *Shenoy et al. (1999)*; (*2002*) included a simulated pursuit condition in their experiments, but the visual stimulus used to simulate pursuit was incorrect. To mimic pursuit, they simply added laminar flow to their fronto-parallel plane (i.e. no local motion parallax cues) optic flow stimuli, and thus their stimuli lacked the dynamic perspective cues necessary to accurately simulate eye rotations on a flat display. When rendering visual stimuli, dynamic perspective cues should be incorporated any time the eye changes orientation relative to the scene (*Kim et al., 2014*).

If eye rotation is simulated (incorrectly) as laminar flow on a flat screen, then it should not be possible for neurons to exhibit rotation-tolerant heading tuning because the addition of laminar motion simply shifts the focus of expansion in the flow field, and does not provide any rotation cues. Indeed, *Bradley et al. (1996)* found that MSTd neurons did not compensate for rotations when pursuit was simulated in this manner. In contrast, *Shenoy et al. (1999)*; (*2002*) reported that MSTd neurons show considerable tolerance to rotation when pursuit was simulated as laminar flow, despite the fact that little or no rotation tolerance was reported psychophysically by the same laboratory for simulated pursuit (*Crowell et al., 1998*). Compared to *Bradley et al. (1996)*, *Shenoy et al. (2002)* used a smaller display size and yet observed larger compensatory effects. This finding contradicts theoretical and psychophysical studies that have established that a larger display size should improve pursuit compensation based on visual cues (*Koenderink and van Doorn, 1987*; *Grigo and Lappe, 1999*). We believe that the counter-intuitive results obtained by *Shenoy et al. (1999)*; (*2002*) stem from the fact that the boundary of their visual stimuli moved across the retina during real and simulated pursuit (but not during the fixation condition), and thus stimulated different regions of the visual field in and around the receptive field of a neuron over time. Such a moving image boundary defined only by the rotation velocity would not occur under natural conditions as a result of eye rotations. By changing the region of visual space that was stimulated over the course of a trial, *Shenoy et al. (1999)*; (*2002*) likely induced changes in the amplitude (response gain) or shape of heading tuning curves.

*Shenoy et al. (1999)*; (*2002*) measured heading tuning over a narrow range (±32°) around straight ahead, and estimated shifts in tuning by cross-correlation analysis. While cross-correlation is invariant to gain changes, it only provides an accurate measure of tuning shifts if the tuning curve has a clear peak within the range of headings tested (*Figure 4—figure supplement 2C,D*; see 'Materials and methods'). In contrast, cells that prefer lateral headings generally have monotonic tuning curves around straight ahead (e.g., *Figure 4—figure supplement 2E*), and this generally yields rather flat cross-correlation functions with no clear peak (e.g., *Figure 4—figure supplement 2F*). As a result, cross-correlation analysis produces fairly accurate estimates of shifts for cells with heading preferences within the range of headings tested, but does not provide reliable shifts for neurons with monotonic tuning functions in that range (*Figure 4—figure supplement 2G*).

These simulations suggest that the degree of rotation compensation reported previously (*Shenoy et al., 1999*, *2002*) may have been inaccurate for neurons with monotonic tuning around straight-forward, which are common in areas MSTd (*Gu et al., 2006*, *2010*) and VIP (*Chen et al., 2011*). This may also help explain the partial rotation compensation observed by *Shenoy et al. (1999)*; (*2002*) in their (incorrect) simulated rotation condition, which contained no relevant visual cues that could be used to compensate for rotation. In contrast to the cross-correlation method, our method for measuring shifts works well for cells with all heading preferences (*Figure 4—figure supplement 1E*), and is robust to variations in the gain, offset and shape of tuning curves.

More recently, *Bremmer et al. (2010)* and *Kaminiarz et al. (2014)* reported that neurons in areas MSTd and VIP, respectively, show rotation-invariant heading tuning based solely on visual cues. However, these studies only measured neural responses to three headings (forward, 30° leftward, and 30° rightward), and defined rotation-tolerance based on a rank-ordering of heading responses across the different eye movement conditions. Since absolute firing rates were not considered, it is likely that shifts in tuning curves could go undetected by this method in the presence of gain fields or bandwidth changes. For instance, this analysis would report identical rank-order for the tuning curves shown in *Figure 4—figure supplement 2A*, and would erroneously classify them as rotation-invariant. In addition, the authors did not attempt to compare their results to the tuning shifts that would be expected if neurons do not compensate for rotation. Consider that, in their ground-plane stimuli (e.g., *Figure 1* of *Kaminiarz et al., 2014*), rotation has a large effect on slow-speed optic flow vectors near the horizon, and high-speed foreground vectors are much less altered. For neurons with receptive fields below the horizontal meridian or those with responses dominated by high speeds, one might not expect the rank ordering of heading responses to change even if neurons do not compensate for rotation. Thus, the results of these studies are difficult to interpret.

By comparison with the above studies, we accurately simulated eye rotations such that correct 2D and 3D visual cues are present in the stimuli. We also measured full heading tuning curves and our analysis methods allowed us to disambiguate changes in response gain from shifts or shape changes in the tuning curve. By using a large display and maintaining the same area of retinal stimulation for all viewing conditions (see 'Materials and methods'), we eliminated artifacts that likely confounded the results of some previous studies (*Shenoy et al., 1999*, *2002*). Therefore, we are confident that our findings in the simulated rotation condition reflect a true contribution of visual cues to the problem of dissociating translations and rotations.

## Implications for self-motion and navigation

In order to navigate through the environment and interact successfully with objects, it is imperative that we distinguish visual motion caused by self-generated movements from that caused by external events in the world (*Probst et al., 1984*; *Wallach, 1987*; *Warren and Saunders, 1995*). For instance, the visual consequences of eye or head rotations need to be discounted in order to accurately perceive whether an object is stationary or moving in the world. The neuroscience literature has extensively studied and emphasized the contribution of efference copy signals to discounting self-generated movements in several sensory systems (*Andersen, 1997*; *Cullen, 2004*; *Klier et al., 2005*). We have presented novel evidence for an alternative solution that is available to the visual system—using large-field visual motion cues to discount self-generated rotations. The ability of VIP neurons to represent heading during rotations, even in the absence of efference copy signals, suggests that visual mechanisms may make substantial contributions to a variety of neural computations that involve estimating and accounting for self-generated rotations.

The contribution of visual cues may be especially important in situations where efference copy signals are either unreliable or absent. For instance, driving along a winding path and looking in the direction of instantaneous heading does not result in any eye or head movements relative to the body (i.e., no efference copy signals). However, such curvilinear motion still introduces rotational components in the optic flow field and disrupts the FOE. In order to estimate such motion trajectories, the visual system would need to decompose self-motion into both translational and rotational components. This study suggests that such trajectory computations based purely on optic flow may be feasible. How the visual system may implement such computations warrants further research and may provide useful insights to neuroscientists as well as those in the fields of computer vision and robotic navigation.

# Materials and methods

## Subjects and surgery

Two adult rhesus monkeys (*Macaca mulatta*), weighing 8–10 kg, were chronically implanted with a circular molded, lightweight plastic ring for head restraint and a scleral coil for monitoring eye movements (see *Gu et al., 2006*; *Fetsch et al., 2007*; *Takahashi et al., 2007* for more detail). Following recovery from surgery, the monkeys were trained to sit head restrained in a primate chair. They were subsequently trained using standard operant conditioning to fixate and pursue a small visual target for liquid rewards, as described below. All surgical materials and methods were approved by the Institutional Animal Care and Use Committees at Washington University and Baylor College of Medicine, and were in accordance with NIH guidelines.

The primate chair was affixed inside a field coil frame (CNC Engineering, Seattle, WA, USA) with a flat display screen in front. The sides and top of the coil frame were covered with a black enclosure that restricted the animals' view to the display screen. A three-chip DLP projector (Christie Digital Mirage 2000, Kitchener, Ontario, Canada) was used to rear-project images onto the $60 \times 60$ cm display screen located ∼30 cm in front of the monkey (thus subtending $90° \times 90°$ of visual angle). Visual stimuli were generated by an OpenGL accelerator board (nVidia Quadro FX 3000G). The display had a pixel resolution of $1280 \times 1024$, 32-bit color depth, and was updated at the same rate as the movement trajectory (60 Hz). Behavioral control and data acquisition were accomplished by custom scripts (see *Source code 1*) written for use with the TEMPO system (Reflective Computing, St. Louis, MO, USA).

## Stimulus and Task

Visual stimuli were presented for a duration of 1500 ms during each trial and consisted of various combinations of eight heading directions in the horizontal plane (*Figure 1C*) and two rotational directions (leftward and rightward). Translation and rotation velocities followed a trapezoidal profile in which the velocity was constant (translation: 24 cm/s, rotation: 17°/s) during the middle 750 ms (*Figure 1D*) of the stimulus period.

The optic flow stimuli were generated using a 3D rendering engine (OpenGL) to accurately simulate combinations of observer translation and rotation. In the 3D cloud protocol, the virtual environment consisted of a cloud of dots that was 150 cm wide, 100 cm tall, 160 cm deep and had a density of 0.002 dots/cm$^3$. The part of the cloud visible to the monkey was clipped in depth to range from 25 cm to 125 cm (relative to the observer) at all times. This clipping ensured that the same volume of dots was visible to the monkey over the duration of a trial as we simulated a translation of 27 cm through the cloud. The stimulus was rendered as a red-green anaglyph that the monkey viewed stereoscopically through red/green filters.

In the second experimental protocol, a fronto-parallel plane (FP) of dots was rendered with a density of 0.2 dots/cm$^2$. The plane was rendered with zero binocular disparity and was viewed by the monkey either binocularly or monocularly, without any red/green filters. During the course of a trial (1500 ms), the 27 cm translation resulted in the simulated distance of the wall changing from 45 cm at the beginning, to 18 cm at the end. We simulated this change in wall distance to better replicate the real world situation of approaching a fronto-parallel wall. Apart from replacing the 3D cloud with a FP plane and the removal of binocular disparity in the stimuli, all other experimental parameters (such as velocity profiles, trial types, stimulus duration, etc) were the same as in the 3D cloud experiment.

During each session, the monkey's eye position was monitored online using the implanted scleral search coil. Only trials in which the monkey's eye remained within a pre-determined eye window (see below) were rewarded with a drop of juice. Trials were aborted if the eye position constraints set by the eye window were violated.

The experiment consisted of three main trial types: pure translation, translation + real eye pursuit (RP), and translation + simulated pursuit (SP). (i) For the pure translation condition, the monkey fixated a visual target at the center of the screen and maintained fixation within a 2° eye window while the optic flow stimuli were presented. Optic flow stimuli simulated eight headings within the horizontal plane, corresponding to all azimuth angles in 45° steps. The pure translation stimuli were rendered by translating the OpenGL camera along one of the eight headings with the velocity profile shown in *Figure 1D*. (ii) For the real pursuit (RP) condition, the animal actively pursued a moving target

while the same translational optic flow stimuli as above were presented on the display screen. A rightward rotation trial started when the fixation target appeared 9.5° to the left of center. Once the monkey fixated this target (within 1000 ms), it moved to the right following a trapezoidal velocity profile (*Figure 1D*). Analogously, leftward pursuit trials began with the target appearing on the right and moving leftward. The monkey was required to pursue the moving visual target and maintain gaze within a 4° eye window during the acceleration and deceleration periods (0:375 ms and 1125:1500 ms). During the middle 750 ms of the trial (constant velocity phase), the monkey was required to maintain gaze within a 2° window around the visual target. Importantly, the optic flow stimulus was windowed with a software rendered aperture that moved simultaneously with the pursuit target. Thus, the area of the retina being stimulated during the RP trials remained constant over time, eliminating potential confounds from moving the stimulus across the receptive field over time (see 'Discussion'). (iii) For the simulated pursuit (SP) condition, optic flow stimuli accurately simulated combinations of the same eight headings with leftward or rightward rotations, while the monkey fixated at the center of the screen (2° window). These stimuli were rendered by translating and rotating the OpenGL camera with the same trapezoidal velocity profile of the moving target in the RP condition. This ensured that the retinal optic flow patterns in the RP and SP conditions were identical (assuming accurate pursuit in the RP condition). The area of retinal stimulation was also identical in the SP and RP conditions.

In addition to these main stimulus conditions, the experimental protocol also included three types of pure rotation conditions for both leftward and rightward directions: (i) eye pursuit over a black background (with the projector on), (ii) eye pursuit over a static field of dots, and (iii) pure rotational optic flow in a 3D cloud (simulated rotation-only). We also included a blank screen during visual fixation and a static field of dots during fixation to measure the spontaneous activity and baseline visual response of the neurons, respectively. Therefore, each block of trials (for both 3D cloud and FP protocols) consisted of 48 unique stimulus conditions: eight directions * (1 translation only +2 RP + 2 SP) + 8 controls.

## Electrophysiological recordings

To record from single neurons extracellularly, tungsten microelectrodes (FHC; tip diameter, 3 μm; impedance, 1–3 MΩ at 1 kHz) were inserted into the cortex through a transdural guide tube, using a hydraulic microdrive. Neural voltage signals were amplified, filtered (400–5000 Hz), discriminated (Plexon Systems), and displayed on SpikeSort software (Plexon systems). The times of occurrence of action potentials and all behavioral events were digitized and recorded with 1 ms resolution. Eye position was monitored online and recorded using the implanted scleral search coil. Raw neural signals were also digitized at a rate of 25 kHz using the Plexon system for off-line spike sorting.

VIP was first identified using MRI scans as described in detail in *Chen et al. (2011)*. Electrode penetrations were then directed to the general area of gray matter around the medial tip of the intraparietal sulcus with the goal of characterizing the entire anterior-posterior extent of area VIP—typically defined as the intraparietal area with directionally selective visual responses (*Colby et al., 1993*; *Duhamel et al., 1998*). To determine direction selectivity, we presented a patch of drifting dots for which the size, position, and velocity could be manipulated manually with a computer mouse. We used this mapping procedure to characterize the presence or absence of strong visual drive as well as the direction and speed selectivity of multi-unit and single-unit activity. At each location along the anterior–posterior axis, we first identified the medial tip of the intraparietal sulcus and then moved laterally until there was no longer a directionally selective visual response in the multi-unit activity.

During each experimental session, we inserted a single microelectrode into the region of cortex identified as VIP. Single unit action potentials were then isolated online using a dual voltage-time window discriminator. Within the region of gray matter identified as VIP, we recorded from any neuron that showed robust visual responses during our search procedure. Once a single unit was isolated, we ran the 3D cloud protocol with all conditions randomly interleaved (72 neurons). Each stimulus was repeated at least four, and usually five, times. At the end of the 3D cloud protocol, if isolation of the neuron remained stable, we ran the fronto-parallel plane (FP) protocol for 4–5 repetitions (34 neurons). For the FP protocol, the red/green

stereo glasses were either removed during the binocular viewing sessions (11/34), or replaced with an eye patch during the monocular viewing sessions (23/34), such that the eye ipsilateral to the recording hemisphere was occluded.

## Analyses

Analysis of spike data and statistical tests were performed using MATLAB (MathWorks). Tuning curves for the different stimulus conditions (translation only, RP, SP) were generated using the average firing rate of the cell (spikes/s) during the middle 750 ms of each successfully completed trial. This analysis window was chosen such that rotation/translation velocities were constant and the monkey was pursuing or fixating the visual target in the small 2° window. To determine the effect of rotations on neural responses, the translation only tuning curve was compared to the RP/SP tuning curves.

Previous studies (*Bradley et al., 1996*; *Page and Duffy, 1999*; *Shenoy et al., 1999*, *2002*; *Zhang et al., 2004*; *Kaminiarz et al., 2014*) only measured tuning curves over a narrow range of headings around straight ahead. Without measuring the full tuning curve, it is very difficult to distinguish between gain fields and shifts in the tuning curves (*Mullette-Gillman et al., 2009*; *Chang and Snyder, 2010*; *Rosenberg and Angelaki, 2014*). Furthermore, these previous studies assumed that rotations would cause a global shift of the tuning curve in the absence of pursuit compensation. However, as shown in *Figure 2* and *Figure 2—figure supplement 1*, rotations can change the shape of the tuning curve, including both skew and bandwidth changes. Therefore, we suspect that the cross-correlation methods or rank-ordering of responses used in previous studies are insufficient to characterize all of the changes in heading tuning due to rotations (see also *Figure 4—figure supplement 2*).

To account for these more complex changes in heading tuning curves, we developed a novel 3-step analysis procedure, as illustrated for an example cell in *Figure 4—figure supplement 1*. Step 1: we measured the minimum and maximum responses of the pure translation tuning curve. The lowest response (trough) and amplitude (maximum—minimum) of the RP/SP tuning curves were then matched to those of the pure translation curve by vertically shifting and scaling the responses, respectively. Step 2: because the predicted effects of rotation are opposite for forward and backward headings (*Figure 2A*), RP and SP tuning curves were split into heading ranges of 0–180° and 180–360°. We tested whether each half of the tuning curve was significantly tuned using an ANOVA ($p \leq 0.05$). All the tuning curves were then linearly interpolated to a resolution of 1°. Step 3: for half-curves that showed significant tuning, we performed a shift analysis as follows. The pure translation tuning curve was circularly shifted (in steps of 1°) to minimize the sum-squared error with each half of the RP/SP tuning curves. For neurons that were significantly tuned in all conditions and in both direction ranges, this analysis yielded four shift values for real pursuit and four shifts for simulated pursuit. In order to quantify the transformation of heading tuning due to rotations, the four shift values were averaged to arrive at one shift value for real pursuit and one shift for simulated pursuit for each cell.

The 95% confidence intervals (CIs) for the shifts plotted in *Figures 4 and 6C*, were calculated using a bootstrap analysis. Bootstrapped tuning curves for translation only, real pursuit, and simulated pursuit were generated by resampling responses with replacement. The same offset, gain and shift calculations were performed on each one of 300 bootstrapped tuning curves to produce a distribution of shifts for each neuron from which the 95% CI was calculated by the percentile method.

In order to test the efficacy of our analysis method, we simulated heading tuning curves using von Mises functions (*Equation 1*), with gain (A), preferred direction (φ), and width (k) as free parameters (*Swindale, 1998*).

$$VM(\theta) = Ae^{\{k[\cos(\theta - \varphi) - 1]\}}. \tag{1}$$

To simulate the tuning curve transformations caused by adding rotational optic flow, a second shape parameter (*Equation 2*) and skew (*Equation 3*) terms were added to the von Mises functions as follows:

$$VM_{width}(\theta) = Ae^{\{k[\cos(\theta - \varphi + \sigma \sin(\theta - \varphi)) - 1]\}}, \tag{2}$$

$$VM_{skew}(\theta) = Ae^{\{k[\cos(\theta - \varphi + \gamma(\cos(\theta - \varphi) - 1) - 1]\}}, \tag{3}$$

where, σ is the second shape parameter (such that slope of the function at half-height can vary independently of the width at half-height) and γ is the skew parameter (see *Swindale, (1998)* for more details). The second shape parameter (σ) was manipulated to yield rotation-added tuning curves with bandwidth changes of 40° (20° on each half of the tuning curve) for cells preferring close to lateral translations ([340°:20°], [160°:200°]). For cells preferring all other headings (close to forward or backward translations), the skew parameter (γ) was manipulated to yield a 20° shift in the peaks of the rotation-added tuning curves.

Random gain values ranging from 0.66 to 1.33 were used to scale the rotation-added tuning curves and random offset values (0–40 spikes/s) were also added to the tuning curves corresponding to leftward and rightward rotations. Poisson random noise was added to all tuning curves (averaged over five simulated stimulus repetitions) and the curves were sampled at heading intervals of 45°, similar to the recorded data. Shifts were measured between the translation only and rotation-added curves using the partial shift analysis method described above. The mean shifts resulting from 10 sets of simulated tuning curves with heading preferences ranging from 0:360° are shown in *Figure 4—figure supplement 1E*. These simulations demonstrate that our method is capable of accurately measuring shifts in the presence of gain, offset and shape changes for neurons with a variety of heading preferences.

To compare our method with the cross-correlation method used in previous studies (*Bradley et al., 1996*; *Shenoy et al., 1999*, *2002*), von Mises functions with Poisson noise were generated as described above (*Equations 1–3*), but were sampled and analyzed as described in those papers. Specifically, simulated tuning curves were generated by sampling the von Mises functions at headings in the range of ±32° around straight ahead, with 8° sampling intervals. To match the previous studies, the resulting data were then smoothed with a three-point moving average and interpolated using a spline function at 1° intervals (*Figure 4—figure supplement 2C,E*). The rotation-added tuning curves were horizontally shifted in 1° increments relative to the translation-only curve and the maximum correlation coefficient between the curves was measured using the equation described in *Shenoy et al. (1999)* (*Figure 4—figure supplement 2D,F*). This analysis was repeated for 10 sets of simulated tuning curves (different random noise samples) for 10 different heading preferences in the range from 0:180° (*Figure 4—figure supplement 2G*). Since this analysis was based only on the narrow heading range of ±32° around straight forward, we did not simulate neurons with backwards heading preferences in the range of 180–360° because such neurons would have little response in this heading range. In contrast with our analysis, this cross-correlation method resulted in unreliable tuning shifts for simulated neurons with heading preferences outside the narrow range of measured headings (*Figure 4—figure supplement 2G*).

To test the rotating reference frame hypothesis (*Figure 7*), the gradient of firing rate was calculated at each heading on each measured tuning curve and the heading associated with the largest positive gradient was selected. The predicted slopes for an eye-centered reference frame were calculated as the average gradient for all the neurons for a given condition (dashed lines in *Figure 7C,D*). To test whether the temporal responses match this prediction, the time course of firing rate was measured at the heading associated with the largest positive gradient, for neurons recorded during the 3D cloud protocol. For sharply tuned neurons, it is possible that the true largest gradient lay between sampled headings. Hence, the measured largest gradient could be part of the peak or trough of the tuning curve. To account for such instances in the data, we excluded tuning curves for which the mean response at the largest gradient heading was not significantly different (t-test; p ≤ 0.05) from the responses of its immediate neighboring headings (29/360 total tuning curves from 72 cells). The time course of firing rate during each trial for the selected heading was calculated by convolving the spike events with a Gaussian kernel (σ = 25 ms). The temporal responses from all selected tuning curves were averaged by condition and used to calculate the mean and standard errors shown in *Figure 7C,D*.

## Acknowledgements

We would like to thank Jing Lin and Johnny Wen for programming assistance and Ari Rosenberg for comments on the manuscript. We would also like to thank Mandy Turner and Tammy Humbird for assistance with animal care.

## Additional information

### Funding

| Funder | Grant reference number | Author |
| --- | --- | --- |
| National Eye Institute (NEI) | NEI, R01-EY-017866 | Dora E Angelaki |
| National Eye Institute (NEI) | NEI, R01-EY-016178 | Gregory C DeAngelis |
| National Eye Institute (NEI) | NEI CORE, EY-001319 | Gregory C DeAngelis |

The funders had no role in study design, data collection and interpretation, or the decision to submit the work for publication.

### Author contributions

AS, Conception and design, Acquisition of data, Analysis and interpretation of data, Drafting or revising the article; GCD, DEA, Conception and design, Analysis and interpretation of data, Drafting or revising the article

### Ethics

Animal experimentation: All surgical and experimental procedures were approved by the Institutional Animal Care and Use Committees at Washington University (#20100230) and Baylor College of Medicine (#AN-5795) and were in strict accordance with the recommendations in the Guide for the Care and Use of Laboratory Animals of the National Institutes of Health.

## Additional files

### Supplementary file

• Source code 1. Custom scripts. Custom scripts written for use with the TEMPO system (Reflective Computing, St. Louis, MO). The scripts define experiment specific parameters and the control loop determining the structure of each individual trial.

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
