## [Decision Letter]

Thank you for sending your work entitled “Role of visual and non-visual cues in constructing a rotation-invariant representation of heading in parietal cortex” for consideration at *eLife*. Your article has been favorably evaluated by Eve Marder (Senior editor), a Reviewing editor, and three peer reviewers.

The Reviewing editor and the reviewers discussed their comments before we reached this decision, and the Reviewing editor has assembled the following comments to help you prepare a revised submission.

Reviewer #1 recommends publication in its current form and has no substantive concerns.

Reviewer #2 had one major concern: The authors make a major compelling case that VIP neurons compensate, at least partially, for rotations, which could allow the monkey to accurately estimate heading in the presences of rotations induced by head or eye movements. The problem is that the degree to which the monkey makes use of these cues is not clear. In an ideal world, the authors could train monkeys to report estimated heading and demonstrate that the monkey compensates, even partially, for rotation induced changes in FOE when it estimates heading (e.g. heading estimates are still pretty good even if made during pursuit). However, this would not be a straightforward experiment to carry out because the changes in FOE induced by pursuit are small, and the monkey's estimates of heading would probably be noisy to begin with. Further, this would entail an entirely new set of experiments that would probably take years to carry out.

According to the reviewer, there two alternatives, as follows:

First, the authors could go into more detail about the psychophysical literature involving heading estimation during pursuit. There are some references to this in the current version of the paper, but the topic is not well-developed.

Second, the authors should further consider ideas about how signals in VIP should be combined to allow an accurate estimate of heading. This would naturally be speculative, but it could provide a very addition to the paper for the following reason: Figure 4 makes clear that neurons vary continuously in the degree to which they are rotation invariant. Indeed, a good handful of neurons (third quadrant) shift in the wrong direction! Of neurons that do shift in the right direction, it is clear that invariance to rotation does not define a category of neurons specialized for this task. Given that rotation invariance is clearly a response feature that is continuously distributed, how might the neurons be combined to allow the kind of heading estimation that subjects actually make? Although the authors won't have behavioral data from these monkeys, they can at least speculate based on behavior in humans.

Reviewer#3: The main result is established thoroughly with appropriate analyses. Two of the subsidiary results are much less convincing and will need additional analyses. First, the claim that VIP can use dynamic perspective cues to detect rotation. She/he is not sure what distinction is here between 2D and 3D. It is true that authors simulate a frontoparallel plan. But it is not true that there is no motion parallax—under perspective projection, the angular distance between points will change with surface motion. How is this not motion parallax? It seems to she/he that the information in dynamic perspective is fundamentally the same as in the 3D cloud, but is much smaller in amplitude and smoother in space. But the distinction is quantitative. So the authors need a clearer statement about what exactly is different.

The other difficulty with this section is that the methods are hard to follow. In the Results, the authors say that the simulated distance of the FP plane changed, but not what the values were. Taken at face value, this would imply a change in the speed of all the dots, which complicates things. But perhaps the simulated translation speed was also changed so that to a first approximation speeds did not change, only the relationship between them? She/he does not understand the reason for doing this experiment with a sudden change at 750 ms. It would be much simpler to do the tuning curves with simulated pursuit in exactly the same way, just using 3D environment that happened to be at a FP plane. Why change everything else?

Second, the discussion of the eye and head centered reference frames is not compelling. Again, part of this is semantic and just needs to be laid out more carefully. In the condition with simulated pursuit, the eye stays stationary. So a truly eye centered co-ordinate frame would imply no changes. Instead, it seems that there is a co-ordinate frame centered on some virtual eye whose position is inferred from the image. But this process of inferring eye position can only infer changes in eye position, so it is hard to see how a center for this reference frame is defined.

A serious issue with the analysis is that the changes rates in Figure 7 only show us that there is some change with eye position. They place no constraints on how closely the co-ordinates follow the eyes rather than the head. Given all the complications they stress elsewhere about the importance of considering the whole tuning curve appropriately, this analysis seems very dangerous. A much simpler approach, that is more robust by the authors' own arguments, would be to construct tuning curves for the first 300 ms of this epoch and the last 300 ms, and then apply the same metrics they develop elsewhere. Finally, the claim that the grey curves in Figure 7 provide control to simple adaptation phenomena is very unclear, because the retinal stimulation s quite different in the pursuit case from the translation alone case. If the condition tested was forward translation (90 degrees), the pursuit stimuli would have substantial net lateral motion, whereas the translation only stimulus would not. So if adaptation primarily affected responses to lateral motion, it could produce exactly this pattern of results.

---

## [Author Response]

*Reviewer #2 had one major concern: The authors make a major compelling case that VIP neurons compensate, at least partially, for rotations, which could allow the monkey to accurately estimate heading in the presences of rotations induced by head or eye movements. The problem is that the degree to which the monkey makes use of these cues is not clear. In an ideal world, the authors could train monkeys to report estimated heading and demonstrate that the monkey compensates, even partially, for rotation induced changes in FOE when it estimates heading (e.g. heading estimates are still pretty good even if made during pursuit). However, this would not be a straightforward experiment to carry out because the changes in FOE induced by pursuit are small, and the monkey's estimates of heading would probably be noisy to begin with. Further, this would entail an entirely new set of experiments that would probably take years to carry out*.

*According to the reviewer, there two alternatives, as follows*:

*First, the authors could go into more detail about the psychophysical literature involving heading estimation during pursuit. There are some references to this in the current version of the paper, but the topic is not well-developed*.

*Second, the authors should further consider ideas about how signals in VIP should be combined to allow an accurate estimate of heading. This would naturally be speculative, but it could provide a very addition to the paper for the following reason:*
Figure 4
*makes clear that neurons vary continuously in the degree to which they are rotation invariant. Indeed, a good handful of neurons (third quadrant) shift in the wrong direction! Of neurons that do shift in the right direction, it is clear that invariance to rotation does not define a category of neurons specialized for this task. Given that rotation invariance is clearly a response feature that is continuously distributed, how might the neurons be combined to allow the kind of heading estimation that subjects actually make? Although the authors won't have behavioral data from these monkeys, they can at least speculate based on behavior in humans*.

Thank you for these suggestions. We are planning to test this behavior in monkeys, but as the reviewer noted, this will be a major endeavor and therefore has to be beyond the scope of this study. However, we hope that the following additions and changes to the paper will address these questions sufficiently.

As per the reviewer’s suggestion, we have added an entire sub-section to the Discussion (‘Comparison to previous behavioral studies’) in which we review the psychophysical literature and compare our results to the existing literature. It is important to note that depending on the parameters of the visual stimuli, psychophysical studies from different laboratories report different amounts of rotation compensation. However, most studies report at least partial compensation during simulated pursuit, and the average compensation exhibited by VIP neurons lies within the range of the behavioral results. This supports our main result that visual cues play an important role in estimating heading during pursuit. In this new section, we also discuss a range of population decoding strategies that could explain the behavioral data based on the neural responses that we observe.

*Reviewer#3: The main result is established thoroughly with appropriate analyses. Two of the subsidiary results are much less convincing and will need additional analyses. First, the claim that VIP can use dynamic perspective cues to detect rotation. She/he is not sure what distinction is here between 2D and 3D. It is true that authors simulate a frontoparallel plan. But it is not true that there is no motion parallax*—*under perspective projection, the angular distance between points will change with surface motion. How is this not motion parallax? It seems to she/he that the information in dynamic perspective is fundamentally the same as in the 3D cloud, but is much smaller in amplitude and smoother in space. But the distinction is quantitative. So the authors need a clearer statement about what exactly is different*.

The motion parallax we refer to in this paper is based on how the concept was first described by Helmholtz (1925) and [21]. If there are two points at different distances from the observer that project to the same retinal location, a translation of the observer results in the closer point having larger retinal velocity than the farther point (assuming the eyes remain still). Rotational optic flow, on the other hand, is invariant to depth. Motion parallax is defined here as the local difference in optic flow vectors between points in the scene that are at different depths but the same retinal location. This has now been clarified in the Results section. Hence, in the case of a frontoparallel plane, such local motion parallax does not exist. To make this distinction clear, we have also rephrased the terminology and now refer to these cues as ‘local motion parallax’ throughout the text.

Rotation of the eye causes global patterns of image motion that indicate that a rotation has occurred. When simulated on a display under planar projection, a pure rotation involves global perspective distortions of the image. The reviewer is therefore correct that such global flow patterns indicating rotation are similar during the 3D cloud and FP plane stimuli. Hence we have modified the text throughout to remove the previous distinction between 3D cues (motion parallax) and 2D cues (perspective distortion).

Importantly, while motion parallax results from observer translations, dynamic perspective cues are a result of eye rotations relative to the scene (under planar projection). This has been explained in detail in a recent paper by one of the co-authors, G.C. DeAngelis (Kim et al., 2014). We emphasize here that dynamic perspective cues are present during both the 3D cloud and frontoparallel plane stimuli. Therefore the specific difference between the 3D cloud and FP plane stimuli is the presence or absence of local motion parallax cues resulting from depth variations. We have added and modified the text to explicitly mention this (please see paragraphs sixteen and seventeen of the Results section).

These edits also include more precise definitions of these cues and clearer explanations of the sources of these cues (i.e., translation vs. rotation). We thank the reviewer for helping us to clarify these issues, which are important.

The other difficulty with this section is that the methods are hard to follow. In the Results, the authors say that the simulated distance of the FP plane changed, but not what the values were. Taken at face value, this would imply a change in the speed of all the dots, which complicates things. But perhaps the simulated translation speed was also changed so that to a first approximation speeds did not change, only the relationship between them?

We have revised the corresponding parts of the Methods to make it easier to follow (‘Stimulus and Task’). The reviewer is correct that the speed of the dots changes as the simulated wall gets closer to the observer. This is because in a real world situation, approaching the wall does in fact result in changes in optic flow speed over time. In contrast, previous studies simulated constant optic flow velocities over the duration of a trial. Such a stimulus corresponds to the impossible scenario in which an observer translates towards a wall, but the distance between the observer and the wall does not change. To us, it makes little sense to simulate this situation. Hence, in order to understand how the brain encodes self-motion, it was necessary that we include these more naturalistic features in our stimulus.

We now describe in more detail this feature of the stimulus in the Methods and explain the rationale and implications of such a stimulus in the Results.

She/he does not understand the reason for doing this experiment with a sudden change at 750 ms. It would be much simpler to do the tuning curves with simulated pursuit in exactly the same way, just using 3D environment that happened to be at a FP plane. Why change everything else?

There was clearly a misunderstanding here. There was no sudden change at 750ms in the FP plane stimulus. The FP plane stimuli were generated using the same 3D environment in which the 3D cloud was simply replaced by a fronto-parallel plane and the disparity was set to zero. The rest of the stimulus parameters, including the velocity profiles, were the same during the two visual scenes. We have revised the Methods to clarify this point.

*Second, the discussion of the eye and head centered reference frames is not compelling. Again, part of this is semantic and just needs to be laid out more carefully. In the condition with simulated pursuit, the eye stays stationary. So a truly eye centered co-ordinate frame would imply no changes. Instead, it seems that there is a co-ordinate frame centered on some virtual eye whose position is inferred from the image. But this process of inferring eye position can only infer changes in eye position, so it is hard to see how a center for this reference frame is defined*.

The optic flow patterns are similar during both real and simulated pursuit and result in the FOE drifting across the retina over time as shown in Figure 6. Hence, in a truly eye-centered co-ordinate frame, we would expect the representation of translation to change over time even in the simulated pursuit conditions. We have clarified this in the Results section.

The reviewer is correct that drift of the FOE over time can be used to infer changes in eye position. We rephrased the relevant sentences in the Results to make this distinction clearer.

*A serious issue with the analysis is that the changes rates in*
Figure 7
*only show us that there is some change with eye position. They place no constraints on how closely the co-ordinates follow the eyes rather than the head*.

This is a very good point. We have addressed this issue by modifying Figure 7 and plotting the expected eye-centered response based on the average tuning properties of the cells (described in the Methods). We find that the slope of measured VIP population responses is significantly different from a slope of 0 and is about 65% of the expected slope based on an eye-centered response. Hence, we can now make a much clearer statement about the extent to which these temporal dynamics are consistent with an eye centered reference frame. The result suggests an intermediate reference frame, but closer to eye-centered than head-centered.

*Given all the complications they stress elsewhere about the importance of considering the whole tuning curve appropriately, this analysis seems very dangerous. A much simpler approach, that is more robust by the authors' own arguments, would be to construct tuning curves for the first 300 ms of this epoch and the last 300 ms, and then apply the same metrics they develop elsewhere*.

The analysis method suggested by the reviewer would not address the question of how the firing rate changes as a result of changing eye position. Measuring the shifts in tuning only provides an indication of which signal is being represented—the translation component or the resultant optic flow. In other words, measuring shifts in tuning provides information about whether a cell compensates for rotations or not. Hence, the suggested analysis would only address how the cells compensate for rotations over time and not about the reference frame in which VIP neurons represent self-motion. In order to evaluate the reference frame, it is necessary to measure the temporal changes in firing, corresponding to changes in the heading relative to the eye. We have added text further clarifying this point (please see the Results section).

*Finally, the claim that the grey curves in*
Figure 7
*provide control to simple adaptation phenomena is very unclear, because the retinal stimulation s quite different in the pursuit case from the translation alone case. If the condition tested was forward translation (90 degrees), the pursuit stimuli would have substantial net lateral motion, whereas the translation only stimulus would not. So if adaptation primarily affected responses to lateral motion, it could produce exactly this pattern of results*.

It is unclear to us why adaptation would primarily affect responses to lateral motion. But even if this were true, we would expect that the firing rate would decrease over time irrespective of the direction of rotation. Hence, this hypothesis would not explain an increase in the firing rate over time for rightward rotations and a decrease for leftward rotations. Moreover, these opposite trends observed during rightward versus leftward pursuit are consistent with the direction expected based on an eye-reference frame hypothesis. We have modified the corresponding section of the Results to focus on the expected outcome of a difference in slope between rightward and leftward pursuit directions, rather than a flat line in the fixation condition. We think that this emphasis largely avoids the problem that the reviewer has raised.